# Polyunsaturated Lipids in the Light-Exposed and Prooxidant Retinal Environment

**DOI:** 10.3390/antiox12030617

**Published:** 2023-03-02

**Authors:** Biancamaria Longoni, Gian Carlo Demontis

**Affiliations:** 1Department of Translational Research and New Technologies in Medicine and Surgery, University of Pisa, 56126 Pisa, Italy; 2Department of Pharmacy, University of Pisa, 56126 Pisa, Italy

**Keywords:** oxidative metabolism, ω3 polyunsaturated fatty acid, elovanoids, xanthophylls, saffron, photoreceptors, retinal pigment epithelial cells, blue light, age-related macular degeneration (AMD)

## Abstract

The retina is an oxidative stress-prone tissue due to high content of polyunsaturated lipids, exposure to visible light stimuli in the 400–480 nm range, and high oxygen availability provided by choroidal capillaries to support oxidative metabolism. Indeed, lipids’ peroxidation and their conversion into reactive species promoting inflammation have been reported and connected to retinal degenerations. Here, we review recent evidence showing how retinal polyunsaturated lipids, in addition to oxidative stress and damage, may counteract the inflammatory response triggered by blue light-activated carotenoid derivatives, enabling long-term retina operation despite its prooxidant environment. These two aspects of retinal polyunsaturated lipids require tight control over their synthesis to avoid overcoming their protective actions by an increase in lipid peroxidation due to oxidative stress. We review emerging evidence on different transcriptional control mechanisms operating in retinal cells to modulate polyunsaturated lipid synthesis over the life span, from the immature to the ageing retina. Finally, we discuss the antioxidant role of food nutrients such as xanthophylls and carotenoids that have been shown to empower retinal cells’ antioxidant responses and counteract the adverse impact of prooxidant stimuli on sight.

## 1. Introduction

Lipids amount to 60% of brain dry weight [1,2] (recently reviewed by [3]) and are enriched in polyunsaturated ω3 docosahexaenoic acid (DHA) (reviewed in [4,5]). In addition, the brain has a high oxidative metabolism: despite representing 2% of body weight [6], its function requires about 20% of the body’s resting metabolic rate, indicating a higher metabolic cost per unit weight than other body tissues. 

The metabolic costs required to restore electrochemical gradients dissipated by spontaneous nerve cell firing and transmitter release show remarkably similar values between different brain areas (discussed in [7]). However, an increase in nerve cell activity above basal level causes the oxygen partial pressure (pO_2_) to increase in the stimulated areas due to enhanced blood flow, with O_2_ influx overcoming consumption. Not surprisingly, given the high lipid content and increased pO_2_ levels in active brain areas, the brain is highly susceptible to oxidative stress damage (recently reviewed by [8,9]). It is important to note that although the increase in pO_2_ associated with increased metabolism occurs in all brain areas, the susceptivity to oxidative stress differs between different brain areas, being higher in the hippocampal, amygdala, and prefrontal cortical neurons (reviewed in [8]). Considering that oxidative stress results from the imbalance between the rates of oxidants and antioxidants generation, differences between different brain areas in their susceptivity to oxidative stress may stem from differences in their regulation of antioxidant levels that relate to the specific functions of oxidants in a given brain area, a notion consistent with a functional role for oxidative stress in the brain, as recently discussed by [10].

In the following, we focus on strategies to counteract oxidative stress, including regulating lipid-derived antioxidants in the retina, a brain-derived area in the eye. The retina has the highest metabolic rates and DHA content in the brain, and oxidant generation is favoured by high pO_2_ levels and light exposure in vivo [11,12] and in vitro [13]. In this oxidative stress-prone brain structure, we discuss the emerging roles of antioxidant systems, including derivatives of very long polyunsaturated fatty acids, xanthophylls and other carotenoids. 

Several recent reviews have covered retinal lipids focusing on their synthesis and antioxidant properties. This review aimed to provide a different perspective on retinal lipids, i.e., their role and antioxidant properties in the anatomical and functional context of rod and cones photoreceptors organisation with pigment epithelial cells and choroidal vessels in a metabolic ecosystem [14]. To this end, we retrieved 28 papers from a PubMed search using the terms DHA and retinal photoreceptors and oxidative stress published between 5 December 2022 and 2000. The selected papers were read, and their relevant references were searched and incorporated in an EndNote X9.3.3 library (Clarivate, Philadelphia, PA, USA) along with additional references from libraries we recently generated for writing reviews and research papers in the area of retinal physiology [15], cell biology [16] and retinal degenerations [15,17]. For Section 6, PubMed searches led to 52 papers on xanthophylls and photoreceptors, three on saffron and photoreceptors and AMD, 29 on vitamin E and AMD and clinical trials as a starting point to discuss roles and mechanisms of action of staple food antioxidants. Overall, we read, evaluated, and discussed the content of over 280 papers, quoting over 200 to provide a critical perspective on retinal lipids in photoreceptors and pigment epithelial cells, considering their anatomy and physiology. 

## 2. The Organisation of Photoreceptors, Pigment Epithelial Cells, and Choroidal Vessels in a Metabolic Ecosystem

The retina, a multi-layered nervous structure in the back of the eye, detects light stimuli via specialised primary sensory neurons, named after their morphologies as retinal rods and cones (see https://webvision.med.utah.edu/book/part-ii-anatomy-and-physiology-of-the-retina/photoreceptors/ (accessed on 24 January 2022)). As shown in Figure 1a, both rods and cones display a spherical cell body (CB) where their nuclei are localised, with an adjacent slender compartment, the inner segment (IS), containing mitochondria highly organised in a non-random position with those of adjacent photoreceptors [18], and the rough endoplasmic reticulum. The IS, in turn, connects to a cilium that provides the pathway for dynamic exchanges between the IS and the outer segment (OS). The OS represents the compartment where light absorption and conversion into an electrical signal occurs via proteins organised in a signal transduction cascade called phototransduction (reviewed in [19]). These compartments are common to rods and cones, although they may differ at the anatomical and functional levels. For instance, the rod OS has a straight appearance and is filled with a stack of membranous disks crowded with proteins of the phototransduction cascade. On the other hand, the cone OS is usually shorter and lacks internal stacked disks.

Upon isolation from the eye, rods and cones in vitro may retain their ability to respond to light for several hours by generating an electrical response. However, their viability and long-term operation require the functional interaction with retinal pigment epithelial (RPE) cells (reviewed by [20]) and a vascular system with peculiar features, the choroidal capillaries (ChC) (reviewed by [21]), whose specific form and shape support its function [22]. The ChC are fenestrated vessels (Figure 1b) of relatively large diameters organised in a dense vascular network, whose high blood flux not only supports the high oxygen consumption by RPE and photoreceptor cells [23,24] but may also prove to be critical for preventing a rise in OS temperature in response to infrared light focusing by the lens [25,26]. The importance of ChC becomes apparent in response to retinal detachment, which may lead to the irreversible loss of photoreceptors [27], or upon faulty ChC development in response to breathing air with increased oxygen partial pressure (pO_2_) by an immature child, as in the retinopathy of prematurity [28]. Last but not least, the RPE also provides the outer retinal blood barrier (RBR), which may restrict access to the retina. Furthermore, ChC and RPE cells also provide the high nutrient fluxes required by the high metabolic rate of photoreceptors [23,24,29,30]. The RPE cells have been conventionally considered supportive in photoreceptors operation, but recent evidence indicates photoreceptors and RPE cells partake in a metabolic ecosystem [14,29,30,31]. 

Although photoreceptors do not generate action potentials and lack the metabolic burden associated with spiking, their oxygen consumption still exceeds that of most stimulated brain areas. The active extrusion of sodium and calcium ions that prevent the dissipation of their electrochemical gradients due to their permeation in darkness [32] (recently reviewed by [33]) accounts for the spiking-independent metabolic cost by oxidative metabolism [29]. Oxygen reaches photoreceptors by diffusion from the ChC: the analysis by O_2_-sensitive microelectrodes indicates pO_2_ decreases to a minimum at about 50 µm from choriocapillaris (see Figure 2), and in vivo measurement of photoreceptor by optical coherence tomography (OCT) [34] indicates that the minimum pO_2_ corresponds to the IS/cell body domains (recently reviewed in p. 19–23 by [16]). Reducing calcium and sodium influx induced by light will reduce oxygen consumption and modifies the pO_2_ spatial profile (Figure 2b). 

In nerve cells, glucose consumption increases more than oxygen consumption [35] (discussed in [7]). The mismatch between glucose and oxygen consumption in response to brain stimulation has been attributed to the glycolytic metabolism required to support glutamate recycling between astrocytes and neurons [35] (recently reviewed by [36]). 

Like the brain, glucose consumption by the retina exceeds that of O_2_. Indeed, despite oxygen availability, retinal glucose metabolism mainly occurs via aerobic glycolysis, as initially reported by Warburg [37]. The use of the less efficient glycolytic metabolism despite oxygen availability appears dictated by the high lipid turnover in retinal photoreceptor cells [38]. Indeed, aerobic glycolysis may contribute a minor fraction of ATP used by rods [39]. Still, photoreceptor cells require a high glucose flux to fuel aerobic glycolysis, which produces metabolic intermediates required for lipid synthesis [14,39,40], matching the high OS turnover in both rods [34] and cones [41]. The high glucose flux required by photoreceptors diffuses from choroidal capillaries (ChC). It must permeate through the glucose transporter GLUT1 across the blood-retinal barrier (BrB) assembled from retinal pigment epithelial cells and their connecting tight junctions [40]. Consequently, mice with reduced Glut1 expression have reduced OS length and a progressively more severe phenotype depending on the extent of Glut1 expression reduction [40]. 

According to the notion of a metabolic ecosystem, RPE cells spare glucose for photoreceptors, which provide RPE cells with lactate to fuel their oxidative metabolism, indicating a symbiotic relationship between photoreceptors and RPE cells rather than an RPE-supporting role [14,39]. Although rod and cone photoreceptors depend on RPE cells to meet their energy requirements, cones require more energy than rods, possibly due to their faster cGMP turnover that accelerates their response to light. Rods support cone metabolism by promoting glucose uptake [42] via the secretion of modified thioredoxin, the rod-derived cone viability factor (RdCVF), which also protects cones from oxidative stress [43]. The mechanism underlying cone cell protection by RdCVF has been attributed to its interaction with the transmembrane protein Basigin-1, which binds GLUT1 and promotes glucose uptake and metabolism by aerobic glycolysis [42]. 

Interestingly, the glucose that supports lipid synthesis for OS turnover by photoreceptors eventually reaches back to RPE cells in the form of membrane lipids. Photoreceptor OS are dynamic structures whose continuous growth must balance the rhythmic shedding of their tips, and RPE cells phagocyte shed OS [44].

In addition to the mutual support between photoreceptors and RPE cells and between rods and cones described above, photoreceptors also provide signals supporting choroidal vessel stability. For example, exposure to increased pO_2_ triggers the retinopathy of prematurity due to impaired choroidal vessel development in premature newborn mice. However, stabilisation of retinal hypoxia-inducible factor (HIF) by the propyl isomerase inhibitor Roxadustat promotes aerobic glycolysis and prevents choroidal vessel loss and retinal damage [28]. 

The overall picture indicates that choroidal capillaries, RPE cells, rods, and cones mutually depend on each other, consistent with the notion of a metabolic ecosystem. However, intrinsic to the ecosystem notion, factors impairing the viability of a member may lead to the functional impairment of the whole system.

## 3. DHA and Oxidative Stress in Photoreceptors and RPE Cells

Most retinal neurons, including photoreceptor cells, do not generate action potentials, and their short axons do not require myelin to improve their cable properties. Analysis of retinal lipid composition indicated phosphatidylcholine (about 40–50%) represents the primary phospholipid, followed by phosphatidylethanolamine (30–35%) and phosphatidylinositol (3–6%) [45]. Overall, membrane phospholipids represent 85% of total lipids in the rat [45] and human retina [46]. Although four main phosphatidylcholines have been found in the mouse retina, they do not show a preferred distribution within the retinal layers [47]. Analysis of retinal Muller glia and ganglion cells indicates their lipidomic profile is consistent with the average retinal lipid content. However, some differences in the relative content of phosphatidylcholine and phosphatidylethanolamine exist between ganglion cells and Muller glial cells [48]. Of note, RPE cells’ lipid content differs from the retina, with a reduced phospholipid content (<60% RPE vs. >85% retina) and an increased content of cholesteryl esters (19% RPE vs. 1.7% retina) [46].

Photoreceptors OS have the highest DHA concentration in the body [49], exceeding the 30% of total lipid in the OS domain of rod photoreceptors [49,50], with rods having higher DHA and very long chain fatty acids (VLCFA) content than cones [51]. The interesting point is that the ω3/ω6 ratio of very long unsaturated fatty acids (≥28 C) and the level of polyunsaturated ω3 and ω6 eicosapentaenoic and arachidonic acids show a significant correlation with plasma and red blood cells ratio, an index of dietary lipids. In contrast, a less strong correlation was observed between retinal and plasma levels of the polyunsaturated ω3 docosahexaenoic acid (DHA) [52]. 

The difference in DHA content between rods and cones may have a functional significance. The rods’ high sensitivity to light allows them to signal the absorption of a single photon and relies on two amplification steps in phototransduction [19]. The first step, the sequential activation of multiple transducin molecules (Figure 3) by one excited rhodopsin, has been estimated to be close to 400 transducins s^−1^ in mammalian rods [53]. Although lower rates have recently been proposed [54] (see however [55]), transducin activation by rhodopsin appears to occur at a substantially higher rate than other GTP-binding protein-coupled receptors. Given the tight packaging of rhodopsin molecules (25,000 µm^−2^) required to attain mammalian rods’ high photon-catching ability, rhodopsin’s high rate of transducing activation must depend on a high diffusion rate in the disk membrane [56]. The diffusion coefficient inversely relates to membrane viscosity, and an increase in membrane fluidity is expected to improve the light sensitivity of rods by increasing the first amplification step in phototransduction.

Rods have upregulated DHA and VLCFA synthesis to increase disk membrane fluidity and the rhodopsin diffusion coefficient. The reduced DHA and VLCFA content of cone OS are coherent with their reduced transducin activation rate and phototransduction amplification step [19,56]. 

The functional benefits of high DHA and VLCFA content come at the expense of an increased risk of oxidative stress in photoreceptors that may trigger retinal degeneration. For example, recent in vivo measurements indicate increased paramagnetic free radicals production in an animal retinopathy model [11], and evidence indicates lipid peroxidation role in light-induced degeneration of retinal photoreceptors [57]. In addition, oxidative stress in RPE cells associated with ageing and environmental factors eventually leads to RPE cells’ demise (reviewed in [58]). This event may also adversely affect photoreceptor cells whose viability requires healthy RPE cells. 

The oxygen gradient along rod OS (see Figure 2 above) and the high DHA content, both have implications for oxidative stress (recently reviewed in [59]) affecting OS lipids and photoreceptor damage [60]. An element to consider in oxidative stress involving photoreceptors and RPE cells is that these cells are exposed to blue light (range 400–480 nm), which may induce oxidative stress [61,62] (reviewed in [63]) via multiple mechanisms (reviewed by [64,65]). Studies using in vitro models have found that blue light-induced oxidative stress in photoreceptors may involve the electron transport chain [66] at the IS mitochondria or specialised OS proteins (opsins) critical for light sensing by photoreceptors [67]. Intriguingly, light-induced oxidative stress and damage that follows OS rhodopsin activation may occur independently of rhodopsin-induced activation of the phototransduction cascade [68]. 

## 4. Vitamin A Derivative All-*Trans*-Retinal and Blue Light may Adversely Affect Photoreceptors and RPE Cells via a DHA-Independent Mechanism

The photoreceptors’ ability to respond to light relies on opsins, specialised proteins that activate in response to photon absorption. Although the peak sensitivity to light of different opsins occurs at different wavelengths of the visible spectrum, all opsins bind the vitamin A derivative 11-*cis* retinal via a Schiff’s base with an amino group. In the retinal-opsin complex, opsin residues help delocalise the electrons of alternating double bonds (conjugated double bonds) of 11-*cis* retinal, shifting its absorption maximum from the UV to the visible spectral region. Upon the absorption of a photon, 11-*cis* retinal isomerises to all-*trans*-retinal (at-RAL), and the activated opsin starts interacting with transducins. Eventually, the at-RAL detaches from the opsin, which deactivates. Once released from opsins, the at-RAL may react with the membrane phospholipid phosphatidyl ethanolamine (PE), forming *N*-retinylidene PE (NRPE). The reaction of an additional at-RAL molecule to NRPE leads to the formation of the phosphatidylethanolamine-bisretinoid (A2-PE), which is then hydrolysed to the photosensitising agent pyridinium bisretinoid *N*-retinylidene-*N*-retinylethanolamine (A2E) [69]. 

The photoreceptors’ strategy to reduce A2E synthesis is to remove NRPE from the cell via a transporter coded by the photoreceptor-specific genes *ABCA4*, which actively transports at-NRPE from the disc lumen to the cytoplasm of rod photoreceptors [70]. In the cytoplasm, phospholipase D hydrolyses NRPE to at-RAL [69], which is then reduced to at-ROL (vitamin A) by retinol dehydrogenases RDH8 and RDH12 [71,72]. The strategy to prevent A2E formation by reducing at-RAL generated in response to light stimuli also prevents A2E generation in response to the 11-*cis*-RAL that fails to bind opsins and reacts with PE [73]. Although both rod and cone photoreceptors of humans, rodents, and porcine retinas express *Abca4*, cones appear to attain higher expression levels than rods, and the expression of ABCA4 protein is also higher in cones than in rods [51,74]. Moreover, cones also have reduced PE levels than cones, and not surprisingly, they also have lower A2-PE levels than rods [51]. Patients with autosomal recessive Stargardt disease type 1 (STGD1) due to pathogenic *ABCA4* variants [74,75] present with macular disease (maculopathy), a clinical finding consistent with the higher turnover rate of cone opsins compared to rhodopsin in rods.

The control over A2E generation and the adverse impact of defects in genes reducing its synthesis is linked to the toxicity of compounds generated from A2E upon exposure to blue light (430 nm) in the presence of O_2_. The A2E has been shown to generate several oxygenated derivatives, such as the nine reactive epoxide rings compound nonaoxirane [76] and superoxide and peroxyl radicals upon irradiation with UV light [77]. From a physiological standpoint, it must be pointed out that eye optics strongly attenuate the UV component of solar light. Therefore, visible blue light should be regarded as the A2E photosensitiser. The A2E-derived products may diffuse from the OS to the IS and then to the cell body where the nucleus is localised, i.e., an anatomical path spanning a distance ranging from 30 to over 150 µm. Assuming for A2E a diffusion coefficient of 1 × 10^−6^ cm^2^ s^−1^ in the viscous cell cytoplasm, its travel time may range from 4.5 to 113 s, indicating that in the absence of scavenging molecules, A2E may affect photoreceptors DNA. However, most damage to photoreceptors likely results from the secondary effects of A2E on RPE cells, where rhodopsin degradation upon phagocytosis of shed OS by RPE cells generates A2E [78]. The RPE cells express ABCA4, although at about 1% of photoreceptors level, at the level of the endolysosomal compartment, where it may remove NRPE generated from 11-*cis*-RAL released in response to the digestion of rhodopsin contained in shed OS phagocyted by RPE cells [79]. In transgenic mice expressing ABCA4 in RPE cells but lacking its expression in photoreceptors, the accumulation of fluorescent A2E was decreased compared to mice lacking ABCA4 in both RPE cells and photoreceptors [79], indicating that a fraction of A2E is indeed generated in RPE cells as a result of rhodopsin digestion. 

Recent data indicate that A2E-derived compounds may adversely affect critical RPEcell components, such as nucleic acids of mitochondria and cell nuclei. Photosensitisation of A2E with 470 nm blue light increases mitochondrial DNA variants, affecting DNA coding regions, including all the oxidative phosphorylation complexes [80], and potentially affecting ATP synthesis. The A2E effects are not limited to mitochondrial DNA, as transcriptome profiling by RNAseq indicates an increase in differentially-expressed genes in pathways involved in angiogenesis, autophagy, extracellular matrix, and cell death in A2E-loaded RPE cells exposed to blue light compared to not exposed RPE cells [81]. Telomere dysfunction with accelerated RPE cell senescence represents an additional impact of photosensitised A2E on the DNA [82]. 

The compound A2E accumulates with time in RPE cells and is a constituent of lipofuscins, fluorescent compounds that accumulate in the back of the eye with ageing. It has been suggested that the accumulation of lipofuscins plays a role in developing age-related macular degeneration (AMD), a disease whose prevalence increases with age. Genome-wide association studies have shown an association of AMD risk with some gene variants, and in particular, the strongest association has been found with genes involved in the complement pathway [83] (discussed in [84]). In vitro experiments indicate an increased conversion of the C3 complement fraction of human serum in inhibited C3b and C3a following the incubation with irradiated ARPE-19 cells, a human RPE cell line [85]. Furthermore, complement activation occurred in an acellular system containing human serum plus A2E oxidised [85], providing support to the notion that lipofuscin component A2E may contribute to low-grade inflammation by activating the complement system and explaining the increased AMD risk associated with gene variants affecting the control of alternative complement pathway (discussed in [86]). A recent study investigated the relevance of complement alternative pathway activation in patients with Stargardt disease caused by ABCA4 mutations (STGD1) but did not find a difference between patients and matched controls [87]. The adverse finding may indicate that in STGD1 patients, RPE cell demise may occur either without complement activation or complement activation may occur locally and could not be assessed at the systemic level in patients. Independent evidence on A2E-induced inflammatory response in RPE cell has been provided in human RPE cells derived from induced pluripotent stem cells (hiPSCs), showing the production of inflammatory cytokines in response to either A2E or oxidative stress with H_2_O_2_ [88].

The impact of A2E-derived compounds upon blue light exposure on the alternative complement pathway may interfere with RPE cells’ role in setting the eye immune privilege via the release of several factors that reduce the reactivity of retina immune cells (recently revised by [89,90]). Furthermore, A2E may affect retinal cell viability via its impact on microglial cells, the primary resident immune cells of the nervous system and the retina (recently reviewed by [91]) that play a role in retinal degenerations caused by genetic defects, as in the case of retinal degeneration 10 mice (Rd10) [92]. Indeed, A2E has been reported to affect microglial cells, promoting their activation, and decreasing complement factor H expression, thus promoting complement activation. Although these effects of A2E on microglial cell activation suggest they may play a synergic role with its direct complement activation [85], it remains unclear how A2E may come in contact with microglial cells in the healthy retina, where microglial cells remain out of the subretinal space. It is possible that A2E uptake by microglial cells does not represent an early event in retinal damage but may instead add up to boost an already ongoing damaging process.

## 5. Lipid-Based Antioxidant Systems in Photoreceptors and RPE Cells

To protect their metabolic ecosystems from oxidative damage, photoreceptors and RPE cells have adopted multiple strategies to manage oxidative stress-induced damage, from developing specialised morphologies to scavenging systems based on specific molecular pathways. 

### 5.1. Photoreceptors’ Anatomy and Eye Structure Afford Protection from Oxidative Damage

In photoreceptors, the sustained oxidative metabolism in darkness may lead to the generation of reactive oxygen species (ROS), such as the hydroxyl ^•^OH^−^ or superoxide O_2_^•−^ radicals [93], and direct measurements in vivo with either newly-developed probes [12] or MRI [11] indicate the IS or the outer retina as the most active retinal site for ROS generation. The ROS may damage sensitive photoreceptor cell components, such as nucleic acids and OS lipids, threatening photoreceptors and RPE cells’ viability. 

The localisation of mitochondria at the IS, a cellular compartment distinct from the nuclear region, and the non-random distribution of mitochondria within the IS [18] may help attenuate the damage expected from highly reactive oxygen radicals, whose travel distance may fall short of the inner segment-nuclear distance. Indeed, the travelling distance of ^•^OH^−^ has been estimated as few Angstroms, while O_2_^•−^ may travel several tenths of microns [94,95], although their charge may prevent these ROS from crossing both plasma and nuclear membranes. This analysis is consistent with the observation that high energy heavy ions (Z ≥ 6) interaction with water may generate ^•^OH^−^, which may interact with lipids to generate peroxyl radicals. The annihilation of two peroxyl radicals leads to the emission of a visible photon (revised in [13]). Astronauts travelling in deep space report the perception of light flashes [96], indicating rhodopsin activation by photons triggered in response to ^•^OH^−^ generation by high energy heavy ions [97]. Consistent with the limited diffusion space of ^•^OH^−^, people on the Earth do not usually report light flashes in darkness, suggesting ^•^OH^−^ generated by oxidative metabolism may not diffuse up to OS, consistent with its limited diffusion space.

Despite the protection afforded by photoreceptor morphology toward genetic material and OS components, photoreceptors need additional factors to protect these sensitive molecular components from the blue light-triggered mitochondrial generation of O_2_^•−^, and lipid peroxidation of DHA and VLCFA, whose travelling distance may extend over IS, OS and nuclear regions. The cornea has high transmittance (around 90%) at wavelengths ranging from 400 to 480 nm, and the energy of sunlight in this spectral region at midday is slightly lower than in the green region of the spectra. Despite the variability linked to daytime, latitude, season, and humidity, the blue/green light ratio stays close to 0.9 at midday [98]. The anatomical organisation of the eye and its optics provide additional protection against blue light-induced oxidant generation. Due to the chromatic aberration of eye optics, blue light is expected to focus at a different axial position from green-red light stimuli. Human eye chromatic aberration analysis indicates an axial focus shift of about one diopter (1D) between 450 and 550 nm light [99]. Considering a 60 D refracting power for the not accommodated emmetropic eye, the −1D axial chromatic aberration will cause blue light to focus about 280 µm in advance of green/red light (550 nm) focus (Figure 4). 

As a result, blue light in the range 415–455 nm, the most effective in generating hydrogen peroxide and the O_2_^•−^ radical in RPE cells [100] and photosensitising A2E in photoreceptors and RPE cells, may still reach these sensitive structures, but its energy will spread over a wider area than longer wavelengths (Figure 4), due to chromatic aberration.

### 5.2. Protection against Oxidative Damage Requires Tight Control over DHA Synthesis

The DHA and DHA-derived molecules provide a second protective mechanism against oxidative damage. Although OS PUFA may undergo peroxidation in response to blue light, emerging evidence indicates that DHA in the retina appears to operate as a double-edged sword, with damaging effects resulting from its oxidation and a protective role due to its metabolism in photoreceptors (reviewed in [101]) and RPE cells ([102,103]). 

The balance between these opposing effects may depend on DHA levels. In the transgenic fat-1 mice expressing a *Caenorhabditis elegans* desaturase able to convert ω6 PUFA into ω3 ([104]) by introducing a double bond, retinal DHA levels increased two to five times in all phospholipid classes compared to WT mice, whereas ω6 fatty acid levels decrease [105]. The increase in DHA also associates with C32 and C34 ω-3 pentaenoic and hexaenoic VLCFA in phosphatidylcholine and depletion of n-6 VLCFAs. From a functional perspective, fat-1 mice have scotopic and photopic ERGs a- (photoreceptor response) and b-wave (bipolar cell response) responses of unusually high amplitudes and lower thresholds, suggesting an increase in OS length [105] and an increased sensitivity to light. The latter effect may reflect an increased membrane fluidity and rhodopsin diffusion coefficient (see Section 3 above). The OS elongation may, however, increase the metabolic burden on photoreceptors due to increased Na^+^ and Ca^2+^ influx leading to increased O_2_ consumption and the development of hypoxic conditions at the IS, causing glial fibrillary acidic protein (GFAP) expression in Müller cells and increased carboxyethylpyrrole (CEP, protein adducts produced from DHA oxidation) in photoreceptors [105]. In mice exposed to intense light, photoreceptors degeneration associated with enhanced lipid peroxidation increased with DHA levels [106], indicating a causal role for DHA in light-induced damage in photoreceptors. The observation of Müller glial cells activation, common during retinal degeneration, suggests that the increase in DHA leading to OS elongation may cause hypoxia and damage in photoreceptors and indicates the need for fine control over DHA levels in photoreceptors to prevent its adverse effects on retinal viability.

Photoreceptors may generate DHA from EPA [107] using the very long fatty acid elongase 2 (ELOVL2) (Figure 5a), which catalyses the conversion of EPA into docosapentaenoic acid (DPA) and then DPA conversion into tetracosapentaenoic acid, which is then converted into the DHA precursor tetracosahexaenoic acid by photoreceptor Δ6 desaturase [107] (reviewed in [4]). 

In mice, controlling DHA synthesis from EPA via the modulation of *Elovl2* expression starts in rod precursors. The *Elovl2* expression is already present two days after birth, i.e., before OS generation, but its levels decrease after postnatal days six, i.e., just before rods start developing an OS. As shown in Figure 5b, the reduction in *Elovl2* expression may depend on the rod-specific transcription factor neural retina leucine zipper (Nrl), as Nrl-KO mice have an increased *Elovl2* expression [108] (see also the RetSeq database at https://retseq.nei.nih.gov/ (accessed on 29 January 2023)). Vitamin A (all-*trans*-retinol) derivatives all-trans and 9-*cis* retinoic acid (RA) promote Nrl expression [109] in photoreceptors. There is evidence that during the early phase of mouse retinal development, both RPE cells and rod precursor express *Aldh1a1*, a gene coding for aldehyde dehydrogenase family 1 subfamily a1 (Figure 5c) to generate at-RA from at-RAL [108] (see also the RetSeq database at https://retseq.nei.nih.gov/ (accessed on 29 January 2023)). 

The orphan receptor transcription factor Nuclear receptor subfamily group E member 3, coded by *Nr2e3* and whose expression is promoted by Nrl, may also contribute to *Elovl2* downregulation [110]. These data may indicate that during the earlier retinal maturation, rods keep *Elovl2* expression high to promote DHA synthesis and boost OS formation. However, rods reduce *Elovl2* expression when OS appears, possibly to avoid their overgrowth that would translate into an increased at-RAL load for RPE cells. It is important to note that also *Aldh1a1* expression drops at about the same time, which would translate into a reduced Nrl activation by at-RA. Reduced Nrl activation by RA would upregulate *Elovl2* expression and DHA synthesis, thus promoting OS growth, and reduced *Aldh1a1* expression may also prevent at-RAL detoxification in at-RA [111]. Recent evidence indicates that in the adult retina, at-RAL detoxification via its conversion into at-RA is carried out by two different enzymes, coded by *Cyp26a1* expressed in Müller glial cells and by *Cyp26b1* expressed in RPE cells [111]. It is relevant that *Cyp26a1* expression in Müller cells starts after postnatal day 4 [112], i.e., when *Aldha1* expression declines in rod precursors, indicating that the two enzymes operate over different temporal windows. 

The finding that rods have lower *Elovl2* expression than cone-like cells of Nrl KO mice [113], while DHA levels are higher in rods than in cones, indicates a mismatch between the expression levels of the gene coding for a critical enzyme in DHA biosynthesis and DHA concentration in rods and cones OS. Although the mismatch cause is unclear, the different DHA turnover between rods and cones may be a possible reason. In rods, DHA is mainly used in phospholipids to increase disk membrane fluidity, increase rhodopsin diffusion coefficient, and amplify the first step in phototransduction (see Section 3 and Figure 3), and its turnover is low. On the other hand, cones may have lower requirements for amplification in their phototransduction. They may instead mainly use DHA as a substrate for generating antioxidant molecules, whose increased turnover requires the higher synthetic rate provided by higher *Elovl2* expression. Although this explanation sounds reasonable, it is important to stress that direct experimental support is lacking.

In keeping with the relevance of control over retinal DHA synthesis, recent evidence indicates additional mechanisms controlling *Elovl2* expression. Adiponectin, a hormone produced by the adipose tissue, regulates insulin sensitivity and glucose levels and has been reported to have antioxidants and anti-inflammatory actions [114]. A reduced *Elovl2* expression in mice lacking both alleles coding for the adiponectin receptor 1 *(Adipor1*) was associated with reduced DHA retinal levels and photoreceptor degeneration by three weeks of age [115,116], indicating the relevance of *Adipor1* in photoreceptors viability. Evidence shows that loss of *Adipor1* in RPE also associates with photoreceptor degeneration [117]. As shown in Figure 5d, Nrl is required to promote *Adipor1*, and the increase in *Adipor1* follows the upregulation of *Nrl* expression (Figure 5e). Therefore, Nrl appears to reduce *Elovl2* expression while promoting *Adipor1* expression, which increases *Elovl2* expression. The Nrl may fine-tune *Elovl2* expression acting via two separate, unknown pathways, although recent evidence points to sphingosine 1-phosphate involvement in signalling via Adiponectin receptors [118]. Adiponectin receptors may not sense adiponectin [119] and instead operate as sensors of membrane fluidity (discussed in [120]), underscoring the importance of controlling membrane fluidity in photoreceptors (see Section 3 and Figure 3 above). 

An additional level of control over DHA synthesis has recently been reported via the increased methylation of the *Elovl2* promoter, which leads to its decreased expression during ageing [121]. In mice, the decreased *Elovl2* expression with ageing may be reversed by intravitreal treatment with 5-Aza-2′-deoxycytidine, which increases *Elovl2* expression and reverses visual function decline [121]. Intriguingly, the *Elovl2* variant C23W disrupting ELOVL2 activity is associated with an early increase in lipofuscin deposition in the mouse retina [121], a sign of early ageing and an indication of DHA in controlling A2E formation.

Figure 5f summarises the multiple control operating on *Elovl2* expression in rod photoreceptors, based on the above evidence, which supports the control of DHA synthesis in photoreceptors at the level of *Elovl2* transcription. However, as shown in Figure 5a, the last step in DHA synthesis requires the translocation of its precursor, tetracosahexahenoic acid, from the smooth endoplasmic reticulum to the peroxisome. Recent data indicate that the selective disruption of the central peroxisomal β-oxidation enzyme in photoreceptors and bipolar cells does not affect photoreceptors’ length and number up to one year of age [122]. As mice lacking this enzyme in all body cells undergo an early loss of photoreceptors, photoreceptors may receive DHA from RPE cells [123], in general agreement with the metabolic ecosystem notion. However, the mechanisms controlling *Elovl2* expression in RPE cells remain unknown (reviewed in [124]).

### 5.3. DHA and DHA-Derived VLCFA Exert Protective Effects toward Oxidative Stress by Turning on Specific Transduction Pathways

In response to oxidative stress generated by either H_2_O_2_ or the herbicide paraquat, able to generate lipid peroxides and other ROS, DHA may activate the ERK/MAPK pathway [125] to prevent caspases activation via the regulation of Bcl-2 and Bax. The ERK/MAPK activation requires the retinoid X receptor (RXR), as shown by the inhibition of DHA protection in the presence of RXR antagonists [126]. The pathway activated by RXR appears specific, as tyrosine receptor kinase (Trk) antagonists do not suppress DHA protection. Moreover, RXR nuclear localisation in rods and the finding that phospholipase A2 inhibition prevents DHA protection suggest that DHA must be released from membrane phospholipids to activate the transcription of specific genes via RXR.

Upon release from membrane phospholipids by phospholipase A2, DHA could be processed by the enzyme 15-lipoxygenase (15-LOX)-1 to generate the protectin 10,17 docosatriene or neuroprotectin D1 (NPD1). This is the stable di-hydroxylated derivative of the short-lived hydroperoxy DHA. The presence of NPD1 has been shown to protect against oxidative stress, inflammation, and apoptosis following stroke by inhibiting cyclooxygenase-2 and the NFkB pathway [127].

Neuroprotectin D1 has been shown to suppress the generation of inflammatory cytokines in response to oxidative stress and the activation of apoptotic mechanisms, indicating a role in counteracting inflammation and apoptosis in photoreceptors and RPE cells (reviewed in [128]). In RPE cells, NPD1 reduces apoptosis triggered by A2E by promoting the expression of members of the Bcl-2 family of antiapoptotic factors and reducing caspase-3 activity in response to oxidative stress [102]. Intriguingly, the neurotrophin Pigment Epithelium Derived Factor (PEDF) promotes NPD1 synthesis and release at the apical membrane of RPE cells facing photoreceptors [102], suggesting NPD1 has both autocrine (to RPE cells) and paracrine (to photoreceptors) actions. The PEDF acts synergistically with DHA to promote NPD1 synthesis and release to protect RPE cells from A2E-induced oxidative stress and damage. In keeping with the notion that photoreceptor and RPE cells organise in a metabolic ecosystem providing mutual benefits, photoreceptor OS phagocytosis by RPE cells has been known to increase their viability, although the underlying mechanisms remained undefined. Both free DHA, i.e., released from membrane phospholipids, and NPD1 increased during OS phagocytosis, suggesting their increase may afford protection against oxidative stress [103]. The notion agrees with the observation of polystyrene microspheres phagocytosis failing to afford neuroprotection against oxidative stress-induced apoptosis and to increase DHA and NPD1. These data suggest that phagocyted OS phospholipids represent the source of DHA required to promote NPD1 synthesis and protect RPE cells and photoreceptors from oxidative damage [103] (reviewed in [101]).

A new class of homeostatic lipid mediators has been identified in the brain as the product of a reaction catalysed by the enzyme elongation of very long-chain fatty acids 4 (ELOVL4) using EPA and DHA as substrates [129], although EPA appears as the preferred substrate [130] (reviewed in [131]); these very long fatty acids (N > 28) are indicated as elovanoids (ELVs) [132]. The first ELVs characterised so far against uncompensated oxidative stress and oxygen/glucose deprivation are ELV-32 and ELV-34 [132]. A difference between DHA and ELVs is their different positions in membrane phospholipids. While DHA forms an ester bond with the alcoholic group in position 2 of membrane phospholipids, such as phosphatidylcholine, the ELV bond occurs at position 1 [132]. 

In a mouse model of Stargardt disease 3 (STGD3) with conditional KO of *Elovl4* in rods and cones [133], the near complete absence of very long chain fatty acids with nearly normal DHA levels did not lead to photoreceptor loss. On the other hand, transgenic mice expressing the mutation causing STGD3 in patients had reduced DHA and very long-chain fatty acid and retinal degeneration. This finding may indicate that the loss of ELOVL4 in photoreceptors does not reduce their viability. Indeed, the adverse effect of the *ELOVL4* mutation associated with STGD3 in patients may result from protein misrouting due to the loss of the dilysine endoplasmic reticulum retention motif [134]. However, human RPE cells (ARPE-19) also express *ELOVL4* and generate Elovs, suggesting they may provide ELV-32 and ELV-34 to photoreceptors lacking ELOVL4 [135]. Application of ELV-32 and ELV-34 to RPE cells exposed to uncompensated oxidative stress (H_2_O_2_) and inflammatory stimuli (TNF-α) protect the cells from apoptotic death by promoting the expression of pro-survival proteins Bcl2 and Bcl-xL, while suppressing pro-apoptotic Bax [135]. The protective action of ELVs appears independent from the increase in NPD1 due to 15-LOX-1 activation by oxidative stress, as ELVs still exert their protection against oxidative stress in the presence of an inhibitor of 15-LOX-1 [135]. The role of ELOVL4-derived elovanoids in photoreceptors and RPE cells has recently been reviewed [136].

## 6. Lipids Endowed with Antioxidant Actions in Staple Foods

Patients with visual loss due to retinal degeneration have few therapeutic options, except for the recently introduced gene therapy for the biallelic pathogenic mutations of RPE65 [137]. Considering the evidence of the role of oxidative stress in triggering photoreceptors and RPE cells’ demise, several nutritional supplements have been investigated in the past to assess their ability to slow down the degeneration and delay the progressive disability. Considering the prevalence of AMD in Caucasians and the ageing population in the western world, most studies have focused on the impact of nutritional supplements on AMD. A 2012 review found that most recent studies at that time focused on DHA, xanthophylls, vitamins B12 and B9, and the active principles of the Age-related Eye Disease Study (AREDS) formulation, i.e., vitamin C, E, beta carotene, and zinc with copper. However, the assessment of specific components of the AREDs formulation by several Cochrane systematic reviews could not find DHA, vitamin E and beta-carotene as beneficial for slowing AMD progression [138,139,140,141,142]. In addition, it was found that beta-carotene in the original formulation increased the risk of lung cancer in smokers and former smokers.

For this reason, the AREDS2 formulation included the xanthophylls lutein and zeaxanthin in place of beta-carotene [143]. A long-term follow-up of the AREDS2 study indicates that 10 mg lutein and 2 mg zeaxanthin were adequate replacements for beta carotene [144]. The evidence for lipidic and non-lipidic antioxidants in AMD has recently been reviewed [145,146]. 

Section 3 and Section 4 have detailed the physiological roles DHA, VLCFA, and aldehydes derived from vitamin A have in vision. Beyond their functional roles in vision, DHA, VLCFA, and vitamin A-derived aldehydes may undergo oxidation, and their oxidized product may have either antioxidant properties or generate oxidants. However, in front of the evidence from in vitro studies and in vivo analysis using animal models, the lack of efficacy for these molecules found in several systematic Cochrane reviews [138,139,140,141,142] is somewhat puzzling. 

In Section 5, we have discussed antioxidant defences, including mechanisms controlling DHA synthesis in photoreceptors, from the early to the late life stages. Such careful control suggests an optimal DHA level so that either lower or higher levels may result in decreased efficacy compared to the optimal level. The notion of optimal DHA level is relevant when considering that data on DHA antioxidant role from in vitro studies and preclinical models do not consider the variable intakes occurring in humans. The point relevance extends beyond DHA and may apply to several lipidic antioxidants in staple foods, such as vitamin E, the stilbene resveratrol, and carotenoids, which have shown antioxidant properties and have been found to protect the retina from degenerating (recently reviewed in [146,147]). In reviewing data from clinical trials on the protection afforded by lipid molecules toward retinal degenerative disease, it is therefore essential to compare the doses tested with their nutritional intakes from staple foods. Accordingly, the analysis must be restricted to those lipid molecules with significant nutritional intakes. In the case of resveratrol, analysis of its plasma levels following intakes consistent with red wine consumption suggests it is only present in trace amounts 30 min from ingestion and is then rapidly metabolized [148]. The average amount of resveratrol contained in red wine (1.9 mg/L) is quite variable (SD = 1.7 mg/L), and the average intake of resveratrol and its glycosides in a Spanish cohort of over 40,000 subjects is less than 1 mg/day (recently reviewed by [149]). For these reasons, we will not analyse in detail the protective role against retinal degeneration afforded by resveratrol as a component of staple food. 

In the following, we will recapitulate the evidence for DHA, xanthophylls, saffron, and vitamin E, focusing on nutritional intakes to address in Section 7 (Discussion) their role as confounding factors that may contribute to the controversial results of clinical trials on their protective role. 

### 6.1. DHA

A 2015 systematic Cochrane review based on two randomized controlled trials (RCT) carried out in France (NAT2), and the USA (AREDS2) compared ω-3 fatty acids to a placebo or no intervention [150]. The review failed to find significant evidence that the treatment either slows AMD progression or reduces the risk of developing moderate to severe visual loss. In the AREDS2 study, patients had either the accumulation of large deposits of lipid material between RPE cells and choroidal capillaries (drusen) in both eyes or large drusen in one eye and advanced AMD in the other eye. In the NAT2 study, patients had early AMD in the study eye and choroidal neovascularization (CNV) in the other eye. The intervention differed slightly. The AREDS2 patients took either 650 mg EPA and 350 mg DHA or the placebo in addition to the AREDS2 formulation with vitamins and zinc. The NAT2 patients received either 840 mg DHA and 270 mg EPA or a placebo. According to the European Food Safety Authority (EFSA), uptake up to 1 g/day does not appear to pose health threats [151] and is about three times the EFSA nutritional recommendation of cumulative EPA + DHA intake (250 mg/day) for the maintenance of normal vision in the general population [152]. The Dietary Reference Intakes (DRI) developed for the USA and Canada recommend ω3 LCPUFA up to 10% of the total energy provided by ω3 PUFA Acceptable Macronutrient Data Range (AMDR) of 0.6–1.2% of daily energy intake. Considering an elderly patient’s average energy intake of roughly 1800 Kcal/day, 1.2% of 2000 Kcal corresponds to 21.6 Kcal, i.e., to 2.4 g of total ω3 PUFA. According to DRI, this patient’s intake would be 10% of 2.4 g, i.e., 240 mg, close to the 250 mg recommended by the EFSA. 

The results from these systematic Cochrane reviews are puzzling, as previous observational studies in 75,889 women of the Nurses’ Health Study (NHS) and 38,961 men of the Health Professionals Follow-up Study (HPFS) older than 50 years found higher intakes of EPA and DHA inversely relate to the risk of developing AMD [153]. Notably, the study assessed EPA and DHA plasma levels to monitor the correspondence between intakes and plasma levels in subjects whose EPA and DHA intakes only come from food to evaluate the impact of long-term nutritional habits. The higher and lower DHA + EPA quintile intakes in the two cohorts were either 416 and 67 for NHS or 697 and 93 mg/day for HPFS, i.e., the intake in the highest quintile was over twice the recommended intake, also due to fish oil supplements taken by 12% of the NHS and 16% of the HPFS group in the upper quintiles. Compared to the lowest quintile of intake, the highest DHA intake reduced the risk of developing intermediate AMD but did not reduce the risk of developing advanced AMD. An interesting finding is that 96% of advanced forms are due to the neovascular form, which is so prevalent over geographical atrophy (GA), the advanced form of dry AMD, suggesting EPA + DHA may prevent more effectively the evolution of dry form to GA.

The efficacy of DHA + EPA in AMD toward the incidence of early AMD was also found in other observational studies, such as the Blue Mountains Eye Study [154]. In the AREDS study, the protection afforded by the highest intakes in ω3 LCPUFA was found to last up to 12 years, as patients with the highest median cumulative intakes of EPA and DHA (160 mg/day) were nearly 30% less likely to progress to advanced AMD [155].

These observational studies indicate that DHA and EPA, as part of a healthy lifestyle, are associated with a reduced risk of developing AMD and progressing from early to intermediate AMD. However, RCT trials could not find a significant reduction in the risk of progressing from intermediate to advanced AMD. 

### 6.2. Xanthophylls

The xanthophyll lutein is a molecule endowed with antioxidant and anti-inflammatory properties that attenuate light-induced oxidative stress, DNA strand breaks, and photoreceptor loss in mice [156] (reviewed in [157]). Lutein is present in the macula, the retinal region with the highest cone photoreceptor density in humans. Chemical investigation of carotenoid derivatives in the macula indicates the presence of zeaxanthin at a higher concentration than lutein [158]; moreover, the macula also contains the stereoisomer meso-zeaxanthin [159]. Interestingly, plasma analysis only found lutein and zeaxanthin, suggesting macular meso-zeaxanthin results from the local conversion of lutein, which has a higher level in plasma than zeaxanthin but lower levels than the two zeaxanthin isomers in the macula [159]. The accumulation of lutein and zeaxanthin in the eye occurs via an uptake mechanism distinct from other carotenoids, such as vitamin A (recently reviewed in [160]). The distribution of xanthophylls in the macula relates to the selective binding of zeaxanthin and meso-zeaxanthin with the Pi isoform of human glutathione *S*-transferase (GSTP1) [161], while lutein binds to the isoform 3 of the steroidogenic acute regulatory domain (StARD3) [162] (recently reviewed in [163,164]). 

Lutein and zeaxanthin accumulate in the macula due to their nutritional intake. Table 1 reports lutein, zeaxanthin, and their cumulated contents for several staple foods from the USDA nutritional database (https://fdc.nal.usda.gov/ (accessed on 15 March 2022)). According to Table 1, spinach, turnip greens, Swiss chard, and collards provide the highest cumulative lutein and zeaxanthin content. Note that one serving size of spinach may provide a cumulative intake of lutein and zeaxanthin nearly equivalent to the supplement provided in three days during the AREDS2 clinical trial (12 mg/day).

It should also be considered that intestinal absorption may affect plasma levels besides food content. Indeed, the bioavailability of carotenoids in vegetables is low (14%), but a substantially higher value has been reported for lutein (67%) [165,166], with several factors, including fat composition and food matrix affecting the intestinal uptake [167].

The relevance of nutritional intakes of lutein and zeaxanthin emerged from observing an inverse relation between vegetable intake and the risk for AMD [168], which prompted the large-scale study Age-Related Eye Disease Study (AREDS2). This study found an inverse relationship between lutein and zeaxanthin intakes and the risk of developing AMD [143]. In a subset of the Rotterdam study population, homozygotes for the *CFH* Y402H allele conferring a high risk of developing AMD had a risk reduction from 2.63 to 1.72, with the lowest risk being associated with the highest tertile intakes (3.23 ± 0.66 mg/day) of lutein/zeaxanthin, while the lowest tertile (1.50 ± 0.25 mg/day) had the highest risk [169]. These findings indicate a significant protective role against early AMD in subjects at higher risk for AMD. Furthermore, analysis of the risk of developing advanced AMD, either wet or non-exudative, in Nurses’ Health Study and Health Professionals Follow-up Study participants indicates that the lutein/zeaxanthin highest quintiles intake (4.78 and 5.47 mg/day, respectively) had an AMD Relative Risk of 0.59 when compared to the lowest quintiles (1.66 and 1.85 mg/day, respectively) [170].

Furthermore, the study indicates an inverse linear relationship between cumulative lutein/zeaxanthin intakes and Relative Risk [170], suggesting that more than 5 and 6 mg/day for women and men could afford higher protection against the risk of developing AMD. Indeed, a linear increase in plasma lutein has been reported in response to lutein intakes up to 20 mg/day [171]. 

Despite the evidence that lutein intakes up to 20 mg/day could be safe, no upper nutritional limits are available for lutein and zeaxanthin [172], and care should be exerted in taking supplements with high dosages that add up with food intake. It should also be considered that plasma levels reach a plateau after about three weeks at this high intake level, suggesting a control over lutein absorption and the lack of benefit for long-term intakes at 20 mg/day or higher. 

An intriguing consideration is that lutein may have protective effects against blue light-induced photoactivation of A2E, a carotenoid derivative, via lutein conversion into meso-zeaxanthin by the isomer hydrolase activity of RPE-65 [173], the critical enzyme in the biosynthesis of 11-*cis* retinal [174], the A2E precursor. Furthermore, the convergence of retinol palmitate and lutein on the same enzyme suggests that RPE65 upregulation may increase both 11-*cis* retinal and meso-zeaxanthin, helping to counteract the adverse effect of 11-*cis* retinal by-product A2E. 

#### Lutein and Zeaxanthin Roles as Blue Light Filters

Section 4 reviewed the evidence indicating that blue light (400–480 nm) affects RPE cells via A2E activation. However, eye anatomy and its optics causing blue light to focus nearly 300 mm from green/yellow light may significantly attenuate its impact in vivo compared to the exposure of cultured cells. Nevertheless, blue light reaches RPE cells, and attenuation may help reduce A2E photoconversion in reactive compounds. Blue light absorption by xanthophylls lutein and zeaxanthin may still help attenuate light reaching RPE cells and prevent oxidative damage in the long term. Both lutein and zeaxanthin absorb blue light with a peak close to 450 when incorporated in liposomes [175]. A randomized, double-blind, placebo-controlled clinical trial in 59 male and female subjects aged 20–69 years found evidence that 16 weeks of diet supplemented with 12 mg/d lutein improved macular pigment density and prevented the loss of resolution due to optical glare [176], an effect related to blue light absorption by xanthophylls rather than by their antioxidant properties. Recently, xanthophylls have been proposed to operate as light-modulated sunscreen with light absorption, causing *trans*-*cis* photoisomerization, promoting their reorientation in the membrane plane and increasing their ability to absorb incident blue light in model membranes as well as in the human retina [177]. Notably, the reorientation occurs on a millisecond time scale, indicating that xanthophylls may operate as fast photodynamic switches. 

The notion of xanthophylls as blue light filters is consistent with their distribution in the primate retina. The analysis indicates that most pigment accumulates in the foveal region but in the outer plexiform layer (OPL) rather than in the IS, where mitochondria operation generates reactive oxygen species or in the OS, where blue light triggers A2E conversion in oxidants [178].

The screening of blue light by lutein and zeaxanthin adds up to the protection provided by chromatic aberration to photoreceptors and RPE cells, but the available evidence indicates that they are not just blue light filters. Indeed, lutein has been reported to reduce ocular inflammation in conditions ranging from endotoxin-induced uveitis to streptozotocin-induced diabetes and retinal ischemia/reperfusion, and in vitro tests have shown its ability to inhibit NFkB activation [157] (reviewed in [179,180]). Moreover, lutein’s anti-inflammatory effects may extend beyond the retina, with reduced activation and decreased secretion of inflammatory cytokines by the BV-2 microglia cell line [181] and reduced inflammatory biomarkers in adults with central obesity [182].

### 6.3. Saffron

Saffron is a spice that has attracted interest in retinal degeneration since the report of the preservation of both anatomy and function in rats prefed with saffron and exposed to bright continuous light [183]. On the other hand, β-carotene prefed rats had anatomical preservation of photoreceptors whose functionality was almost lost [183]. Although this initial observation was intriguing, defining the underlying molecular steps involved in the mechanisms of food requires the definition of its chemical constituents as well as the possible role of cultivars, soil characteristics and climatic properties on active components. In the case of saffron, more than 150 different chemicals have been extracted, but only a third have been identified. Chemical analysis indicates that saffron protective effects on retinal degeneration may relate to its crocin content [184,185], a group of esters between crocetin (a diterpenoid and natural carotenoid dicarboxylic acid) and disaccharides, such as gentobiose. Although crocin and crocetin have antioxidant properties, the protection against light damage in rats suggests they may act via specific pathways. Continuous light upregulates the expression of cannabinoid receptors type 1 and 2 (*Cbr1* and *Cbr2*) transcripts and their protein products [186] with a more substantial effect on CBR2. The protein CBR2 has been linked to immune system modulation (reviewed in [187]) and reported to be upregulated in cultured human Müller glial cells isolated from cadaveric donors in response to bacterial lipopolysaccharide (LPS) [188]. The LPS also promoted the transcription of several inflammatory cytokines, such as TNF-α and IL-6, and the endocannabinoid 2-arachidonoylglicerol (2-AG) inhibited their transcription and release in the medium acting through the mitogen-activated protein kinases (MAPK) [188] suggesting that activation of cannabinoid receptors may reduce LPS-triggered inflammation. The observations that saffron blocks CB1R and CB2R upregulation in response to continuous light and the protection from retinal damage in response to continuous light by the intravitreal injection of either a selective CBR1 antagonist (rimonabant) or a CBR2 inverse agonist (SR144528) [186], suggest that saffron may act on retinal target distinct from Müller glial cells. Analysis of ERG’s response to light stimuli found an increase in the a-wave response at the highest intensities tested, indicating an increase in the response of photoreceptors in *Cbr2* KO mice compared to wt [189]. Although these findings suggest that blocking CBR2 may protect photoreceptors, the acute application of the inverse agonist AM-630 did fail to increase ERG a-wave, while a seven-day application mimicked the increased responses observed in *Cbr2* KO mice [189]. The authors concluded that the effects of *Cbr2* KO may represent an effect not directly linked to the block of signal transduction via the CB2R, as shown by the inverse agonist AM-630 failing to affect the ERG response. 

Another critical point with food is assessing whether the effects observed in vitro or animal models could be replicated in patients. A comparison of saffron and lutein/zeaxanthin supplements in AMD patients indicated that sight remained stable in saffron-treated (20 mg/day—https://clinicaltrials.gov/ct2/show/record/NCT00951288 (accessed on 14 January 2023)) and declined in lutein/zeaxanthin-treated patients over 29 ± 5 months [190]. In a randomized and controlled clinical trial, Stargardt patients with defects in *ABCA4* (see Section 4 above) were treated for six months with saffron supplements (20 mg/die) or a placebo and assessed at baseline and after six months. Vision did not decline in patients treated with saffron, while a tendency toward a decrease was observed in patients treated with a placebo [191]. Two reviews of different studies in patients treated with saffron concluded that saffron has a promising potential as an effective and safe adjunct therapy [192,193] in AMD [194,195,196,197,198], glaucoma [199] and diabetic retinopathy [200]. These data suggest that 20 mg/day of saffron may slow down the time course of visual loss in AMD patients, and it is, therefore, relevant to compare saffron nutritional intakes. A list of over 30 recipes using saffron (https://www.bbcgoodfood.com/recipes/collection/saffron-recipes (accessed on 26 February 2023)) indicates this spice is common to Spain, Italy, India and several other countries in South Europe and Middle-East. The saffron amount for rice and saffron recipe provides about 50 mg/serving indicating the daily doses administered to patients fall in the range provided by three serving/week. A point that deserves attention is that the study in patients [190,191] used a patented saffron formulation (Saffron Repron), and its match with saffron produced from different cultivars raised in areas with soil and climate differences is presently unclear.

### 6.4. Vitamin E

Tocopherols are a class of lipid-soluble molecules with varying degrees of antioxidant activity. Among tocopherols, α-tocopherol is considered the most active form and representative of vitamin E activity. Vitamin E is a potent antioxidant with anti-inflammatory activity, reducing lipid peroxidation and superoxide ion O_2_^•−^ production as well as inflammatory cytokines production and the inflammatory marker reactive C protein (reviewed in [201]).

Considering the high content of ω3 LCPUFA in photoreceptors OS and the elevated pO_2_ in the subretinal space where OS sit (see Section 2 and Section 3), it is reasonable to expect α-tocopherol would protect photoreceptors and RPE from oxidative stress and inflammation associated with retinal degenerative diseases. The AREDS and the AREDS2 studies included 400 IU of vitamin E and 500 mg of vitamin C in their formula. One IU vitamin E corresponds to 0.67 mg of α-tocopherol (https://dsid.od.nih.gov/Conversions.php (accessed on 27 February 2023)), indicating AREDS and AREDS2 provide a daily intake of 268 mg of vitamin E equivalent to 268 mg of α-tocopherol. This amount should be compared to α-tocopherol RDA for adults provided by DRI (https://nap.nationalacademies.org/read/9810/chapter/1 (accessed on 27 February 2023)) (15 mg/day for both genders) and AI provided by EFSA (13 and 11 mg/day for men and women, respectively) [202]. Although the chronic vitamin E intake of patients enrolled in AREDS and AREDS2 studies is over 15 times the RDA, the DRI also reports the tolerable upper level (UL) for α-tocopherols at 1000 mg/day, indicating that the high intake provided by the treatment may not pose a health threat to enrolled patients.

Although both AREDS studies found that including the high dose of vitamin E reduced the progression from intermediate to advanced AMD, studies analysing the impact of vitamin E alone did not provide compelling evidence for its efficacy. In a prospective randomised, placebo-controlled clinical trial in 1193 healthy volunteers aged between 55 and 80 years, 500 IU vitamin E daily for four years reduced neither the incidence of early AMD compared to placebo (8.6% vs. 8.1%) nor the development of late AMD (0.8% vs. 0.6%) [203]. In a randomized, double-masked, placebo-controlled trial, 14,236 healthy male physicians aged ≥ 50 received 400 IU vitamin E or placebo on alternate days and 500 mg vitamin C or placebo daily for eight years. The results indicate no significant differences associated with treatment, with 96 and 97 AMD cases in the vitamin E and placebo groups, respectively [204]. A Cochrane systematic review of eight randomized trials compared vitamins (vitamin A or beta-carotene) or mineral supplements to assess their efficacy in preventing AMD or slowing its progression over a treatment time of 4–12 years. The results indicated no evidence for vitamin E or beta-carotene in preventing AMD. However, combining vitamins and minerals slowed AMD progression to advanced AMD and visual acuity loss, suggesting that antioxidant vitamin supplements do not prevent AMD [138,205]. Two additional Cochrane systematic reviews have analysed the impact of vitamin E on the development [141] or the progression [142] of AMD using a more extensive database. The conclusion was that vitamin E did not prevent AMD or slow its onset and that people taking antioxidant vitamins were less likely to progress to late AMD, although the overall effect was modest. 

## 7. Discussion

High oxygen availability provided by choroidal capillaries to support oxidative metabolism [33], exposure to visible light stimuli in the range of 400–480 nm, and high polyunsaturated lipids content [49,50] make the retina an oxidative stress-prone tissue [13]. Indeed, lipid peroxidation and their conversion in reactive species promoting inflammation have been reported and connected to retinal degenerations [11,12,57,58]. However, despite these adverse conditions, the retina provides us with visual insights over several tens of years, resulting from a strategy based on multiple layers of protection. Besides the first level, based on the anatomical organization of the retina to avoid reactive species generation close to sensitive cell structures, the organization in a metabolic ecosystem with the release of neurotrophin such as RdCVF [42,43] affords a second level of protection. Finally, multiple and independent evidence indicate a third level of protection afforded by retinal lipids oxidation that generates lipid mediators [128], such as NPD1 [101,102,103], counteracting the inflammation and death pathways triggered by specific signalling pathways in response to signals from the prooxidant retinal environment.

More recently, novel very-long fatty acids, longer than 28 carbon atoms, have been identified [132] and chemically characterized as products of elongase *Elovl4*, coded by a gene whose defects have been found cause retinal degeneration in patients. These elovanoids appear to have protective actions independent from NPD1 [5,135,136].

The fourth level of protection is afforded by lifestyle, with nutrients found in staple foods as increased levels of their intake having proved protective against some forms of retinal degeneration. It is important to note that for the AREDS2 formulation, including the xanthophylls lutein and zeaxanthin, the National Eye Institute has clarified that the formula does not prevent early AMD from developing into intermediate AMD but may prevent intermediate AMD from developing into late AMD (see https://www.nei.nih.gov/learn-about-eye-health/eye-conditions-and-diseases/age-related-macular-degeneration/nutritional-supplements-age-related-macular-degeneration (accessed on 19 January 2023)).

This clinical observation may indicate that multiple mechanisms operate in retinal degeneration and that the efficacy of a given formula may only prove effective over mechanisms operating in a specific stage of the clinical course. A related concept is that different chemicals may operate on different mechanisms over different stages of the disease, possibly explaining why saffron has been reported to be more effective than lutein and zeaxanthin in delaying AMD progression over a relatively short assessment time [190]. 

In discussing the pleiotropic actions on retinal cells viability and function of several lipids, such as ω3 LCPUFA and carotenoids, their antioxidant properties provide the starting point. Two recent reviews have analysed the discrepancies between the antioxidant properties measured in vitro and the results from clinical trials [206,207]. A relevant case is that of β-carotene, whose potent antioxidant actions and scavenging of singlet oxygen may predict a protective role against cancer, while clinical trials found an increased risk of cancer in smokers [208,209]. As discussed in [206], β-carotene antioxidant properties may depend on the interaction with other food components, and it may exhibit good radical trapping properties for pO_2_ tension lower than those in normal air (150 Torr or about 0.2 Atm) [210]. Indeed, at pO_2_ levels close to normal air, β-carotene may lose its antioxidant properties and, especially at higher concentrations, may display autocatalytic prooxidant behaviour [210]. 

The context-dependent switch from anti- to prooxidant behaviour may also hold for other carotenoids with conjugated double bonds, a notion relevant as we have reviewed in Section 2 the evidence of elevated pO_2_ levels in RPE cells. Indeed, analysis in the human retina found that xanthophylls effectively quench singlet oxygen generated in response to white light stimuli [211]. However, the study was conducted in the macula of cadaveric donors, and the pO_2_ was likely lower than in vivo. In fact, in a heterogeneous lipid/water environment, the efficacy of β-carotene and zeaxanthin in preventing unsaturated fatty acid methyl esters peroxidation increased with concentration and was inversely related to pO_2_ and did not show a clear dependence on carotenoids structural properties [212]. Zeaxanthin’s antioxidant properties toward egg yolk phosphatidylcholine appear enhanced by its interaction with the binding protein GSTP1 [213]. Considering accumulation of zeaxanthin in the OPL, it seems unlikely that it may exert significant antioxidant actions in the OS and RPE cells.

Furthermore, the similar antioxidant properties of β-carotene and xanthophylls may not match the protection afforded by xanthophylls over β-carotene toward AMD progression from intermediate to advanced AMD [214]. Furthermore, crocin, the aglycone of the saffron pigment and active principle crocetin, has been reported to lack scavenging activity against the superoxide anion displayed by xanthophylls [215]. However, saffron appears more effective in AMD patients than xanthophylls [190]. Last, the lack of evidence from clinical trials that the chain-breaking antioxidant vitamin E does not afford protection against AMD or its progression from early to advanced AMD suggests that the antioxidant properties of several lipid molecules may not fully account for their protection toward the progression from intermediate to late AMD. 

As recently suggested for carotenoid-mediated health benefits [207], carotenoids and ω3-LCPUFA may have pleiotropic actions via cellular signalling pathways controlling the expression of genes coding for enzymes involved in the cellular responses to antioxidants. In some cases, they may synergize [216], possibly by converging on retinoid receptors [126]. Lutein has recently been proposed to act in mice via multiple antioxidant pathways in response to light-induced oxidative stress [217], reducing oxidant generation and, in parallel, promoting the expression of superoxide dismutase coding genes *Sod1* and *Sod2* and reducing markers of macrophages recruitment to the RPE-choroid. A recent review covers antioxidant enzymatic pathways that protect photoreceptors and RPE cells from degeneration [218].

The notion of retinal degeneration as a multiple-stage process involving stage-specific mechanisms, ranging from oxidative stress through inflammatory responses, may help explain the puzzling findings provided by data from clinical trials assessing the protection by lipid molecules in staple foods toward AMD. Most observational clinical trials enrolling many patients, conducted over 6–12 years, and monitoring long-term nutritional intakes of ω3 LCPUFA and xanthophylls provided evidence for protection against incident AMD [153,154,155]. On the other hand, systematic Cochrane reviews generated negative results for ω3 LCPUFA preventing the progression from intermediate to advanced AMD [150]. Although randomized double-blind and placebo-controlled clinical trials represent the gold standard in clinical research, and the evidence they provide is considered superior to observational trials, it is also apparent that in this case, they have been looking for different outcomes, i.e., the prevention of incident AMD or the progression from intermediate to advanced forms. These different outcomes may result from different mechanisms operating in the initial stages, where DHA and other LCPUFA may play a protective role, and in the progression from intermediate to advanced forms, where the ω3 LCPUFA are not effective. The DHA accumulates in photoreceptors, and substantially lower levels are found in RPE cells. Daily intakes of 250 mg EPA + DHA may afford protection against blue light induced A2E conversion to epoxides, sparing an oxidative burden on RPE cells phagocyting shed OS, thus protecting RPE cells from blue light photosensitization. However, once lipofuscin accumulates in RPE cells, ω3 LCPUFA in photoreceptors may not prevent the progression from intermediate to advanced AMD that involves the RPE cells. Similar consideration may apply to xanthophylls, which have low levels in RPE cells and accumulate in the OPL layer.

An important concept emerging from discovering multiple protective mechanisms is the need for their coordination. We have reported and discussed the emerging evidence for multiple control levels setting the expression of *Elovl2*, which codes for a critical enzyme in the conversion of EPA into DPA and in the further processing of DPA in the pathway to DHA. Furthermore, multiple transcription factors control *Elovl2* expression during development, adult life, and ageing, but the mechanisms controlling these transitions and the consequence of their faulty operation remains unassessed. An area that will require additional investigation is the definition of pathways affecting gene coding for the generation of protective lipid derivatives and the response to oxidants, as they may provide insights into strategies for improving retinal cell viability.

The methodological advances in single-cell transcriptomics may provide novel insights into the molecular organization of cell ecosystems, such as that describing the interactions between rods, cone and RPE cells, with significant advances in the understanding at the molecular level of their advantages in terms of resilience and the role lipid derivatives play in a complex organization of multiple cell types. 

## 8. Conclusions

An important concept emerging from discovering multiple protective mechanisms is the need for their coordination. We have reported and discussed the evidence for multiple control levels ranging from the anatomical organization of photoreceptors and RPE cells, the distribution of macular xanthophylls, and DHA synthesis in photoreceptors. The different distributions of these molecules in photoreceptors and RPE cells may explain their protection in different stages of retinal degenerative diseases. 

## Figures and Tables

**Figure 1 antioxidants-12-00617-f001:**
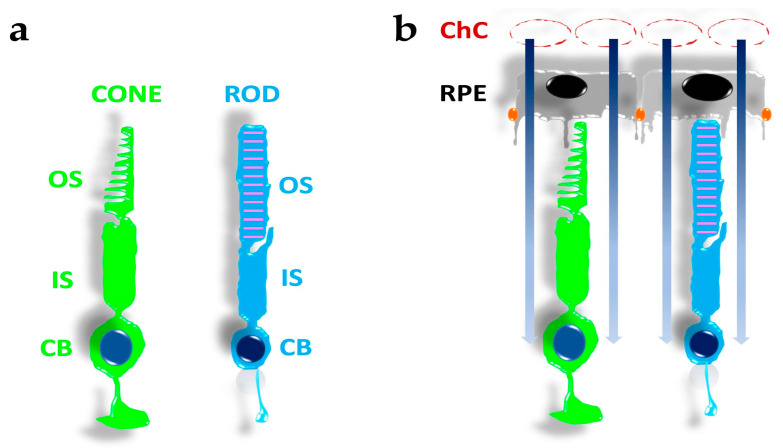
(**a**) Schematic representation of a cone (green) and a rod (cyan) photoreceptor, with outer segment (OS), inner segment (IS), cell body (CB); (**b**) Choroidal capillaries (red) (ChC) provide O_2_ flux (downward pointing arrows) to retinal pigment epithelial (RPE) cells (grey) and cone and rod photoreceptors.

**Figure 2 antioxidants-12-00617-f002:**
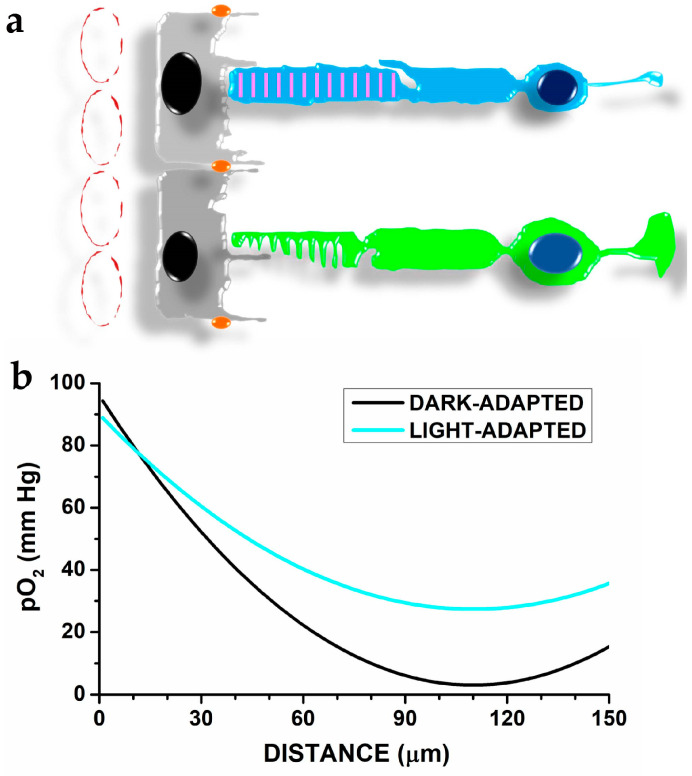
(**a**) Schematic representation of the anatomical relationship between ChC (as red color), RPE cells (as black color) and photoreceptors; (**b**) The black line plots the computed pO2 extracellular profile going from ChC to the cell body of photoreceptors.

**Figure 3 antioxidants-12-00617-f003:**
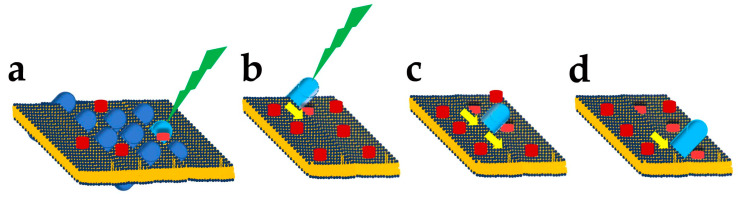
(**a**) Schematic representation of membrane phospholipid bilayer with hydrophilic heads (dark blue) screening hydrophobic lipid tails (yellow) from water. Integral membrane protein rhodopsin (blue) and membrane-associated transducin (dark red). Upon light stimulation (green), rhodopsin became active (cyan) and turned on transducin (light red); (**b**–**d**) Upon activation by light (**b**), one rhodopsin activates several transducins (**b**,**c**) while diffusing (yellow arrows) in the membrane.

**Figure 4 antioxidants-12-00617-f004:**
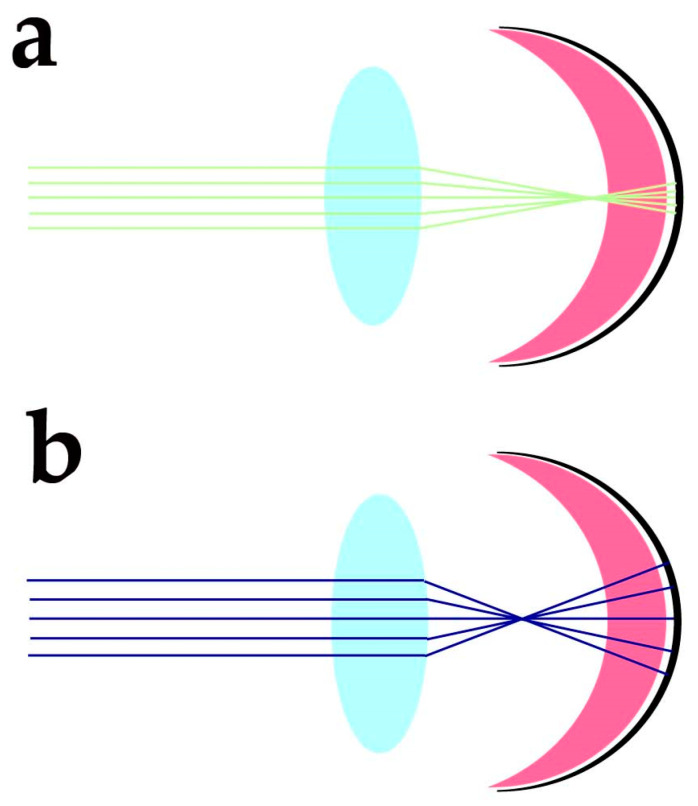
(**a**) Stray yellow lines represent green-yellow light being refracted by the eye optics represented by the cyan lens, with the image formed at the level of photoreceptors OS in the subretinal space; (**b**) blue lines represent blue light being refracted by the cyan lens, with the image formed at the level of photoreceptors OS in the subretinal space. The different focus positions for blue and green-yellow light indicate the chromatic aberration of the eye optics. As a result of chromatic aberration, the blue light will spread over a wider area of RPE than the green-yellow light.

**Figure 5 antioxidants-12-00617-f005:**
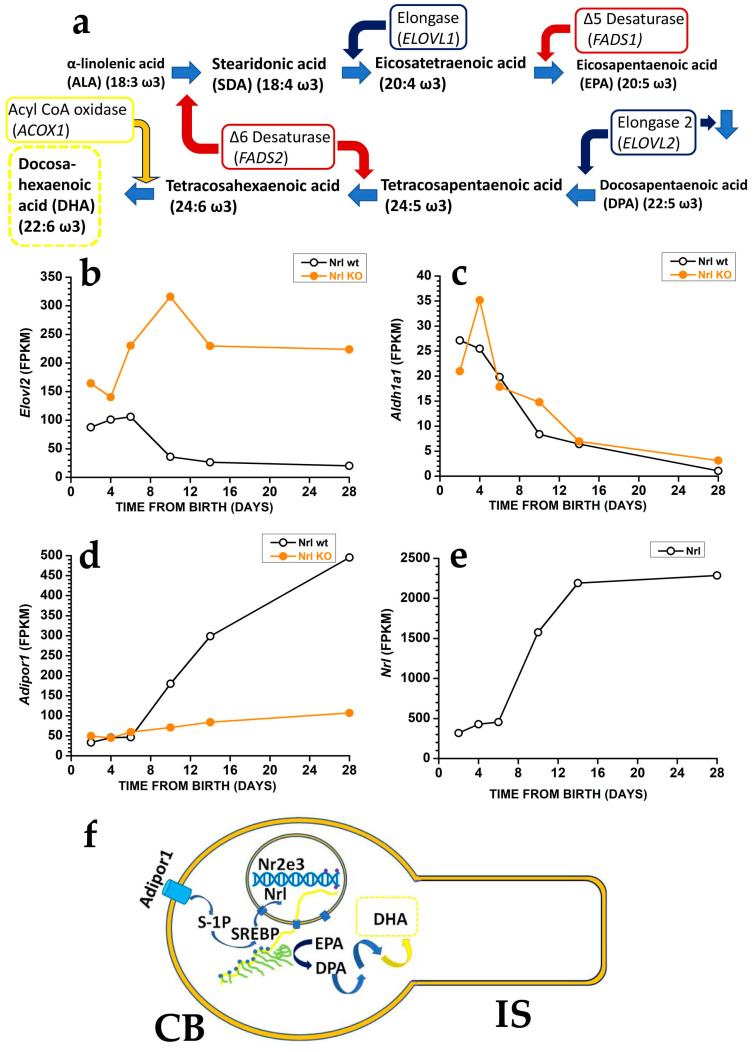
(**a**) Enzymatic pathway converting essential fatty acid alpha-linolenic acid to EPA and DHA. Elongases are enclosed in blue squares, and desaturases are in red squares. Note that ELOVL2 converts EPA into DPA. The yellow boxes indicate the peroxisomal compartment, where tetracosahexaenoic acid conversion into DHA by ACOX1 occurs; (**b**–**d**) Circles plot expression in rod precursors as fragments per kilobase exon per million fragments (FPKM) at different times from birth of wt (black circles) and Nrl-KO (orange circles) mice for (**b**) *Elovl2*; (**c**) *Aldh1a1*; (**d**) *Adipor1*; (**e**) Circles plot Nrl expression in rod precursors in FPKM at different times from birth of wt mice; (**f**) Scheme summarising the control over *Elovl2* transcript (yellow line crossing the nuclear membrane) and protein (green) by nuclear transcription factors Nrl and Nr2e3, by the transmembrane protein Adipor1 (cyan transmembrane cylinder) via sphingosine-1 phosphate (S-1P) and the sterol regulatory element binding protein1 SREBP1, and by DNA methylation (purple circles). Panel data (b-e) have been redrawn using values downloaded from the RetSeq database at https://retseq.nei.nih.gov/ (accessed on 29 January 2023).

**Table 1 antioxidants-12-00617-t001:** Lutein and zeaxanthin content of staple food from the USDA database.

FOOD	Serving Weight (g)	Serving Size	Lutein (µg)	Zeaxanthin (µg)	Lutein + Zeaxanthin (µg)
Grade A egg	50.3	1 Egg	116	115	
Tomatoes, grapes, raw	49.7	5 Tomatoes	47.2	4.47	
Butter stick, salted	100	100 g	12	4	
Milk, whole, 3,25% fat	249	1 Cup	12.4	2.49	
Milk, non-fat	246	1 Cup	2.46	0	
Spinach, frozen, cooked, no fat	215	1 Cup	---	---	33,600
Spice, paprika	2.3	1 TSP	---	---	436
Cress, raw	50	1 Cup	---	---	6250
Turnip greens, frozen, cooked, drained	220	1 Package	---	---	26,200
Swiss chard, frozen, cooked, drained	175	1 Cup, chopped	---	---	19,300
Collards, frozen, cooked, no fat	175	1 Cup	---	---	18,500
Radicchio, raw	40	1 Cup	---	---	3530
Kale, fresh, cooked, no fat	130	1 Cup	---	---	8290
Peas, green, cooked, boiled, drained	160	1 Cup	---	---	4150

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
