# Peer review of "Polyunsaturated Lipids in the Light-Exposed and Prooxidant Retinal Environment"

_antioxidants, 2023, doi:10.3390/antiox12030617_

Round 1
Reviewer 1 Report
The manuscript by Longoni and Demontis is a comprehensive, up to date and interesting review on the importance of polynsaturated lipids and of oxidative stress in the retinal environment.
Given the pathogenic role of oxidative stress and the possibility to implement antioxidant approaches to ameliorate retinal diseases, I would suggest to slightly expand chapter 6. At present Authors have focused their attention to xhanthophylls and saffron. They did not even mentioned studies and clinical trials for instance on resveratrol, zinc, omega-3. See for instance two recent reviews by Wang (Clin Ophthalmol 2021, 15: 1621-1628) or by Dziedziak (Antioxidants 202, 10: 1743-1767).
Author Response
POINT 1: Given the pathogenic role of oxidative stress and the possibility to implement antioxidant approaches to ameliorate retinal diseases, I would suggest to slightly expand chapter 6. At present Authors have focused their attention to xhanthophylls and saffron. They did not even mentioned studies and clinical trials for instance on resveratrol, zinc, omega-3. See for instance two recent reviews by Wang (Clin Ophthalmol 2021, 15: 1621-1628) or by Dziedziak (Antioxidants 202, 10: 1743-1767).
REPLY: In the revised version, section 6 discusses several lipid antioxidants found in staple foods, such as vitamin E, resveratrol, and DHA. Considering that the section title is lipid antioxidants in staple food, we present evidence indicating that resveratrol from food (mainly wine) only reaches trace amounts in plasma for about 30 minutes after intake. In the revised section 6, DHA (6.1) and Vitamin E (6.4) have been included and discussed. Also included is the evaluation of their efficacy in AMD from clinical trials has been added. The recent reviews suggested by reviewer #1 fit nicely in the expanded section 6 and are now quoted as [146] and [147].
Reviewer 2 Report
The topic of the present review is relevant for understanding the particularity of biochemical processes and regulation and functions of lipids and lipids-derived antioxidants (carotenoids) in the retina, a tissue prone to high oxidative stress due to numerous factors.
The authors have carried out a comprehensive analysis on the topic of unsaturated lipids (mainly DHA) and other molecules which are involved in the oxidative metabolism at the level of photoreceptors and RPE. The manuscript represents a critical, constructive analysis of the literature in the field, covering aspects related mainly to the description of the specific environment or “metabolic ecosystem” of the retina (as called by the authors), underlying the double role of very long chain fatty acids (oxidation substrates/source of protective compounds by metabolism) and the involvement of natural dietary antioxidants. All the aspects are presented from an anatomical perspective, which in my opinion is an original and valuable approach. It does not only present an overview of relevant streams of thought in the topic covered, but also adds new insights on developments in the area, indicating what the open questions are.
The manuscript covers the most relevant literature in the field (both historical and recent), it is clearly written and well organized. For me it was a real pleasure to read this manuscript. The tables and the figures are adequate in number and content.
Some suggestions and comments are presented below:
Suggestions:
1. The authors should include some information about applied methods, data sources (e.g. bibliographic databases), search terms and search strategies, selection criteria, the number of studies screened and the number of studies included etc.
2. A brief presentation of the lipid composition (types of lipids, fatty acids profile) in different structures of the retina would be useful to complete the picture.
3. Tocopherols are also natural lipophilic antioxidants, present in retinal structures in much higher concentration than in plasma. Even more, the AREDS trials included a large amount of tocopherol (400 mg) in both formulas, a lot above the RDA. In my opinion it would worth to address this topic.
4. Beside the blue light filter properties of macular xanthophyll’s, they are also acting as antioxidants in the retina. I suggest to include a brief presentation of the most relevant studies demonstrating this action. Additionally, in this section the authors should include the specific binding proteins, responsible for the capture and deposition of macular carotenoids (tubulin, pi isoform glutathione S-transferase, e.g.) – see recommended literature below.
I would also suggest the introduction of the following relevant references for carotenoids:
1. Mechanistic aspects of carotenoid health benefits - where are we now? Bohn, T; Bonet, ML; (...); Dulinska-Litewka, J, Dec 2021 | Nutrition Research Reviews, 34 (2), pp.276-302
2. From carotenoid intake to carotenoid blood and tissue concentrations - implications for dietary intake recommendations, Bohm, V; Lietz, G; (...); Bohn, T, May 2021 | Nutrition Reviews, 79 (5), pp.544-573
3. Mechanisms of Transport and Delivery of Vitamin A and Carotenoids to the Retinal Pigment Epithelium, Earl H Harrison, Mol Nutr Food Res, 2019 Aug;63(15):e1801046. doi: 10.1002/mnfr.201801046.
Minor comments and corrections
Lines 53-54: Xanthophylls are also carotenoids, so I suggest changing with “xanthophylls and other carotenoids”, or simply carotenoids
Line 212: Retinal is not a conjugated ”diene”, as it contains 5 conjugated double bonds. I suggest changing with conjugate system or conjugated double bonds.
Line 219: Maybe it would useful to ads here that A2E act as a photosensitizer, as a reason for the necessity to reduce its synthesis, which is described further
Line 349: in the Title of Figure b) blue lines represent light being refracted by the cyan, I think “blue” light is missing
Line 390: In the figure 5 please replace “stearadonic” with “stearidonic” acid an a-linolenic wit alpha-linoleic (in the title of the figure)
Line 560: In the title of the section correct “xhanthophylls” with “xanthophylls”.
Line 608: mg/day instead of mg/die
Author Response
MAJOR POINTS:
Some suggestions and comments are presented below:
Suggestions:
- The authors should include some information about applied methods, data sources (e.g. bibliographic databases), search terms and search strategies, selection criteria, the number of studies screened and the number of studies included etc.
In designing the review, we realized that merging data from developments in long-chain n-3 fatty acids chemistry with recent cell biology data on RNA-seq and placing it in a physiological context may provide a novel perspective in the field. To this end, we retrieved 28 papers from a PubMed search using the terms DHA and retinal photoreceptors and oxidative stress from December 5th, 2022, to 2000. The selected papers were read, and their relevant references were searched and incorporated in an EndNote X9.3.3 library along with additional references from libraries we recently generated for writing reviews and research papers in the area of retinal physiology, cell biology and retinal degenerations. For section 6.0, PubMed searches led to 52 papers on xanthophylls and photoreceptors, three on saffron and photoreceptors and AMD, 29 on vitamin E and AMD and clinical trials as a starting point to discuss roles and mechanisms of action of staple food antioxidants. Overall, we read, evaluated, and discussed the content of over 280 papers, quoting over 200 to provide a critical perspective on retinal lipids in photoreceptors and pigment epithelial cells, considering their anatomy and physiology.
In the revised version, the last paragraph of Section 1 (l. 55-70) summarizes the approach followed to generate a critical and updated review of retinal lipids in the retinal context.
- A brief presentation of the lipid composition (types of lipids, fatty acids profile) in different structures of the retina would be useful to complete the picture.
REPLY: In the revised version, section 3 (l. 171-191) now discusses the lipid composition of RPE cells, retina, retinal glial cells, ganglion cells and photoreceptors and indicates that retinal DHA levels correlate with its plasma and RBC levels (indexes of dietary lipids), while a highly significant correlation exists for retinal EPA and AA and their plasma and RBC levels. Likewise, a significant correlation exists between ω3/ω6 ratios of very long-chain PUFA in the retina and plasma or RBC.
- Tocopherols are also natural lipophilic antioxidants, present in retinal structures in much higher concentration than in plasma. Even more, the AREDS trials included a large amount of tocopherol (400 mg) in both formulas, a lot above the RDA. In my opinion it would worth to address this topic.
REPLY: In the revised version, Section 6.4 now reports on studies investigating Vitamin E in AMD and retinal viability (l.820-860).
- Beside the blue light filter properties of macular xanthophyll’s, they are also acting as antioxidants in the retina. I suggest to include a brief presentation of the most relevant studies demonstrating this action. Additionally, in this section the authors should include the specific binding proteins, responsible for the capture and deposition of macular carotenoids (tubulin, pi isoform glutathione S-transferase, e.g.) – see recommended literature below.
I would also suggest the introduction of the following relevant references for carotenoids:
- Mechanistic aspects of carotenoid health benefits - where are we now? Bohn, T; Bonet, ML; (...); Dulinska-Litewka, J, Dec 2021 | Nutrition Research Reviews, 34 (2), pp.276-302 (quoted p. 21, l. 930)
- From carotenoid intake to carotenoid blood and tissue concentrations - implications for dietary intake recommendations, Bohm, V; Lietz, G; (...); Bohn, T, May 2021 | Nutrition Reviews, 79 (5), pp.544-573 (quoted p. 16, l. 684)
- Mechanisms of Transport and Delivery of Vitamin A and Carotenoids to the Retinal Pigment Epithelium, Earl H Harrison, Mol Nutr Food Res, 2019 Aug;63(15):e1801046. doi: 10.1002/mnfr.201801046.(quoted p. 16, l. 681)
REPLY: In the original version, we mentioned that xanthophylls might act as antioxidants in the retina. In the first sentence of section 6.1, we stated: “The xanthophyll lutein is a molecule endowed with antioxidant and anti-inflammatory properties that attenuate light-induced oxidative stress, DNA strand breaks, and photoreceptor loss in mice)”. In the revised version, the antioxidant actions of carotenoids have been addressed in the discussion (section 7.0), where we discussed their role as antioxidants in the context of the retinal organization (l. 895-929). In the revised version, we also quote the updated references on the protective role of macular xanthophylls suggested by the reviewer [207] [163] [160].
The uptake from plasma and binding proteins are now detailed in section 6.2 (l.679-684), In the revised version, section 6.2.1 now includes data indicating xanthophylls' impact on vision unrelated to their antioxidant actions (l. 742-753).
The discussion now analyzes retinal lipids as having pleiotropic action, via their scavenging properties as well as via signalling pathways (l. 930-962).
Minor comments and corrections
Lines 53-54: Xanthophylls are also carotenoids, so I suggest changing with “xanthophylls and other carotenoids”, or simply carotenoids
Line 212: Retinal is not a conjugated ”diene”, as it contains 5 conjugated double bonds. I suggest changing with conjugate system or conjugated double bonds.
Line 219: Maybe it would useful to ads here that A2E act as a photosensitizer, as a reason for the necessity to reduce its synthesis, which is described further
Line 349: in the Title of Figure b) blue lines represent light being refracted by the cyan, I think “blue” light is missing
Line 390: In the figure 5 please replace “stearadonic” with “stearidonic” acid an a-linolenic wit alpha-linoleic (in the title of the figure)
Line 560: In the title of the section correct “xhanthophylls” with “xanthophylls”.
Line 608: mg/day instead of mg/die
REPLY: all the minor points listed above have been fixed in the revised version.